# Analysis of the Contribution of Intrinsic Disorder in Shaping Potyvirus Genetic Diversity

**DOI:** 10.3390/v14091959

**Published:** 2022-09-03

**Authors:** Guillaume Lafforgue, Thierry Michon, Justine Charon

**Affiliations:** 1UMR Biologie du Fruit et Pathologie, INRAE, Université de Bordeaux, CS 20032, 33882 Villenave d’Ornon, France; 2Marie Bashir Institute for Infectious Disease and Biosecurity, School of Life and Environmental Sciences and School of Medical Sciences, The University of Sydney, Sydney, NSW 2006, Australia

**Keywords:** protein intrinsic disorder, potyvirus, diversity, mutational robustness

## Abstract

Intrinsically disordered regions (IDRs) are abundant in the proteome of RNA viruses. The multifunctional properties of these regions are widely documented and their structural flexibility is associated with the low constraint in their amino acid positions. Therefore, from an evolutionary stand point, these regions could have a greater propensity to accumulate non-synonymous mutations (NS) than highly structured regions (ORs, or ‘ordered regions’). To address this hypothesis, we compared the distribution of non-synonymous mutations (NS), which we relate here to mutational robustness, in IDRs and ORs in the genome of potyviruses, a major genus of plant viruses. For this purpose, a simulation model was built and used to distinguish a possible selection phenomenon in the biological datasets from randomly generated mutations. We analyzed several short-term experimental evolution datasets. An analysis was also performed on the natural diversity of three different species of potyviruses reflecting their long-term evolution. We observed that the mutational robustness of IDRs is significantly higher than that of ORs. Moreover, the substitutions in the ORs are very constrained by the conservation of the physico-chemical properties of the amino acids. This feature is not found in the IDRs where the substitutions tend to be more random. This reflects the weak structural constraints in these regions, wherein an amino acid polymorphism is naturally conserved. In the course of evolution, potyvirus IDRs and ORs follow different evolutive paths with respect to their mutational robustness. These results have forced the authors to consider the hypothesis that IDRs and their associated amino acid polymorphism could constitute a potential adaptive reservoir.

## 1. Introduction

### Protein Intrinsic Disorder

Proteins possess intrinsically disordered regions (IDRs), i.e., regions lacking a unique three dimensional structure and yet capable of exerting important biological functions [1,2], which challenges the so-called “structure-function relationship” dogma. Although it is currently quite widely admitted that intrinsically disordered hub proteins are key players in the cellular interactome, the involvement of intrinsic disorder (ID) in evolution is still under debate. An earlier study was aimed at comparing the structural features of single-domain small proteins from hypothermophylic bacteria, archaea, mesophilic eukaryota and prokaryota, and RNA or DNA viruses, whose crystal structures were available [3]. It was concluded from this analysis that viral proteins, and more particularly RNA virus proteins, display: (i) higher stability upon simulations of mutation accumulation; and (ii) lower inter-residues contact densities (i.e., mean number of interactions—van der Waals interactions and hydrogen bonds—per amino acid residue) [4]. This latter feature is a typical signature of intrinsic disorder. It has thus been proposed that the large intrinsic disorder content in viral proteins could help to efficiently buffer mutation effects [3,5]. This was experimentally shown in the case of the intrinsically disordered protein VPg from certain potyviruses [6]. This contrasts strongly with the non-additive/epistatic stability loss profile expected from ordered proteins as previously reported for a bacterial β-lactamase [7]. It is therefore conceivable that low structural requirements in IDRs could lead to a capacity to accumulate amino acid substitutions without drastic impairment of protein function. For instance, this idea sounds especially relevant regarding RNA virus adaptation to hosts. This could contribute to rapid adaptation to environmental stresses without an excessive loss of fitness. There is no doubt that this question is of general interest to the virology community. Consequently, the high evolutionary potential of RNA viruses, and the high ID content in their proteins, has set the basis for assessing the contribution of ID to the shaping of virus genetic diversity in a context of host adaptation. Plant-phytovirus pathosystems provide useful experimental models for studying these aspects [8].

An in silico analysis unveiled a high ID content in the *Potyvirus* proteome both at inter- and intra-species scales [9]. This feature has been conserved during *Potyvirus* evolution, suggesting the functional advantage of ID. When comparing the evolutionary constraint (ratio of non-synonymous to synonymous substitution rates, dN/dS) between ordered and disordered regions within the proteome of different potyvirus species, IDRs display significantly higher dN/dS values than ordered regions (ORs), a finding that indicates the tendency of intrinsically disordered domains to evolve faster than more structured regions during potyvirus evolution [9]. Using the pathosystem PVY/pepper, we previously obtained the first in vivo experimental data supporting the hypothesis that IDRs could influence virus adaptability to the host [10], possibly by enabling a faster exploration of the mutational space and thereby allowing the virus to bypass the plant’s resistance. Indeed, a correlation was observed between the adaptive potential of the virus and the disorder content within the VPg viral protein.

To further assess this previously described role of IDRs on RNA virus adaptation, the present study aimed at analyzing whether the regions predicted as disordered in viral proteomes are more likely to evolve and accommodate amino acid substitutions (non-synonymous mutations) than more structured areas. In this work, we define mutational robustness as the propensity of a given genomic region to accumulate amino acid substitutions. Whether the mutations are deleterious or not in certain hosts or environmental conditions remains to be evaluated. Ordered and disordered region sequences from various potyvirus species were thus retrieved and compared for several adaptive parameters at two timescales of viral evolution, namely a short -term scale experimental evolution and a long-term evolution reflected by natural diversity. 

The short-term scale data analyzed consisted in high-throughput sequencing (HTS) retrieved from three independent evolution experiments (i.e., PVY [11,12], and TEV [13]). HTS provides access to the complete genome sequences of all viral variants—including those that are in a minority—that make up a population [14]. By sequencing each individual genome from the viral population, it is thus possible to assess the genetic structures of evolving potyvirus populations and thus potentially address the processes that shape this genetic variability (and, to a greater extent, the evolvability of the viral population (i.e., a virus ability to increase its fitness over time in the course of an evolution process)).

To evaluate the impact of disordered versus ordered regions on potyvirus proteins-mutational robustness at a higher scale of evolution, genomic sequences from TuMV, TEV, and PVY’s natural diversity were also retrieved.

To prevent bias in our analysis of the structural determinant of potyvirus evolution, a third dataset corresponding to simulated data was also obtained. Briefly, potyvirus genomes were artificially mutated in silico according to their viral replicase features to mimic the genetic diversity obtained in the absence of natural complex phenomena shaping the virus mutational landscape such as selection, genetic drift, or complementation within the viral population.

Adaptive parameters of the resulting mutants were thus obtained and compared to those from the biological data.

## 2. Material and Methods

### Data Sets

○Disorder prediction

We scanned for disordered regions along potyvirus polyproteins using the ‘predictor of naturally disordered regions’ (PONDR-VLXT), an algorithm accessible through the server (http://www.pondr.com (1 September 2022) [15,16]. Parameters were set to “default” for ID score predictions.

○Experimental dataset

The study of Cuevas et al. 2015 (TEV 2015), [13] evolved the TEV (*Tobacco etch virus*) on two different host, *Nicotiana tabacum* and *Capsicum annuum*. The mutations occurring within both hosts after passages 1, 4, 6, 9, 12, and 15 were pooled and compared to the initial TEV strain. For the two others studies, namely Kutnjak et al. 2015, 2017 [11,12] (PVY 2015 and PVY 2017), using the PVY (*Potato virus Y*) on *Solanum tuberosum*, mutations were compiled for a single three weeks residence and for passages from 1 to 6, respectively. They were compared to the initial strain. Appendix A compiles all of the resulting mutations of experimental datasets.

○Natural diversity dataset

Datasets used contained 6 genomes of TEV isolates, 100 genomes of PVY isolates, and 100 genomes of TuMV (*Turnip mosaic virus*) isolates. Corresponding genome accessions are listed in Appendix A. These datasets will be referred to as TEV_ND_, PVY_ND_ and TuMV_ND_ in the study.

○Simulation

The distribution of mutations in the virus sequence results from from the sum of the contribution of viral polymerase errors and the subsequent complex interplay between selection, genetic drift, and possible complementation between variants which cannot be simply discriminated. In order to uncouple these two components, we built an algorithm to mimic the distribution of synonymous and non-synonymous mutations introduced by the low fidelity virus RNA polymerase during genome replication (DOI: 10.5281/zenodo.6396239). It was hypothesized that mutations could be randomly introduced all along the genome during its replication. Consequently, if IDRs and ORs were equally susceptible to mutations, NS and S were expected to be homogenously distributed in each of the two regions before virus submission to the selection pressure and other evolutionary forces. The simulation also takes into consideration the specificity of viral polymerase on transversion/transition mutations calculated from TEV experimental data [17].

We generated 1000 variants from the original potyvirus sequence, with each variant bearing one SNP.

The genome sequences used for simulations were as follows: GenBank accession numbers TuMV: D83184, PVY: AJ439544.2, and TEV: DQ986288.1

○Adaptive components tested

The collections of sequences generated from experimental evolution, natural diversity, and simulated experiments where then analyzed with respect to the number of S and NS mutations (DOI: 10.5281/zenodo.6396239) and for each viral protein, their location either in the ORs or IDRs. BLOSUM-based scores of each NS mutations were also used to determine the potential of IDRs and ORs to cope with amino acid substitutions (DOI: 10.5281/zenodo.6396239). Finally, the characteristic of naturally occurring substitutions were analyzed in term of maintenance versus disturbance of disorder. It was reported that ORs and IDRs possess distinct sequence biases. Disorder-promoting scores, ranging from 0 to 1 (1 being the highest disorder-promoting score for disorder) were adapted from a previously published classification [18] and associated with each amino acids (Appendix A).

○Codon volatility assessment

Codon volatility values, corresponding to the 61 sense codons under the simplest mutation model, were retrieved from [19] and used to calculate the mean volatility for the three TuMV, PVY and TEV reference genomes using a script previously published in [6]. Codon volatility values obtained for IDRs and ORs were compared within each viral species using a non-parametric Mann-Whitney U-test and mean values. For each viral species, simulated codon volatility data were also obtained by generating 1000 random nucleotide sequences encoding the same amino acid sequence but varying by one base at each codon (see 10.5281/zenodo.7032281 for the corresponding script). The mean codon volatilities of each variant randomly generated were assessed as described previously and compared to the values obtained for the corresponding viral species. 

## 3. Results

To evaluate the contribution of intrinsically disordered regions on potyvirus evolvability, this study compared viral genomic populations retrieved from evolution experiments, natural diversity, and in silico generated pool of variants. Several parameters were thus assessed and compared at both genomic and proteomic levels, consisting of: (i) the location of the diversity (within intrinsically disordered versus ordered protein regions); (ii) the nature of nucleotide mutations (synonymous versus non-synonymous); and (iii) the biochemical and disorder-promoting nature of the corresponding amino acid substitutions. 

### 3.1. Theoretical Minimum Number of Mutations Required for an Accurate Estimation of S and NS Distribution in the Genome

Datasets generated from the 3 experimental evolutions represent between 115 and 317 mutations. We hypothesized that the number of mutations considered could be too low to lead to robust conclusions. Consequently, our first consideration was to estimate the average number of mutations required to be significant. A total of 4 independent generations of 100, 300, 500, 750, 1000 and 1250 mutations were thus randomly introduced along the TEV genome sequence. Assuming that such random mutagenesis should not be impacted by any structural or protein determinants, the number of mutations (synonymous and non-synonymous) should be equally distributed along the genome, regardless of the corresponding proteome intrinsic disorder. Thus, the distribution of NS and S mutations among IDRs were determined (Figure 1). Above 600 mutations, an equal distribution of mutations among either ORs and IDRs was observed. Therefore, in order to ensure representative values for further analysis, the results of 4 independent simulations with 1000 mutations each will be used.

This threshold of 1000 mutations, which is required for a robust analysis, was confirmed by monitoring the evolution of the R^2^ correlation coefficient between the number of mutations and viral protein length (see the following subsection) as a function of the mutation number. Whether for NS or S, below 750 mutations, the R^2^ greatly fluctuates (Appendix A).

This result confirms that the limited number of mutations available from the experimental datasets is likely to make our analysis less robust. To increase the size of the dataset and extend our observations to larger scales of viral evolution, the natural diversity of TEV_ND_, PVY_ND_, and TuMV_ND_ isolates was also analyzed by retrieving complete genomes available in Genbank. We identified 1296, 4646, and 7528 mutations in the corresponding TEV_ND_, PVY_ND_ and TuMV_ND_ datasets. These data should allow us to assess whether there is a significant difference in mutational robustness between IDRs and ORs.

### 3.2. Correlation Assessment between Protein Length and Number of Mutations

Several factors can influence protein propensity to accumulate amino acid substitutions. One of them could be intrinsic disorder, which is highly heterogeneous between potyviral proteins [9]. To investigate the link between ID and mutational robustness at the protein level, we first assessed the propensity of each potyvirus protein to accumulate non-synonymous (NS) versus synonymous (S) mutations. The percentage of NS or S mutations observed in each protein coding sequence was calculated as a function of the total amount of NS or S mutations in the genome for each of the experimental evolution, natural diversity, and simulated datasets (Figure 2).

At the short-term evolution scale, the longer the protein, the higher the number of S mutations, with a significant correlation between protein length and percentage of S mutations. By contrast, NS mutation numbers were not correlated with the protein length, for the three experimental studies analyzed [11,12,13] (Table 1).

At the long-term evolution scale, the natural diversity confirmed the trend that the accumulation of NS mutations poorly correlated with protein length. Non-synonymous mutations, potentially adaptive and reflecting the viral amino acid polymorphism, are thus not accumulated homogeneously along the potyvirus proteome.

Regarding the simulated data, S and NS mutations are equally represented along potyvirus mutated genomes independently of the protein length (R^2^ = 0.94), thus validating our random model for a number of mutations above 1000. As expected, it is indicative of the correlation between the protein sequence length and the number of S or NS mutations obtained at random in the absence of any biological bias.

It should be noted that for all simulated data, the number of NS mutations is three times higher than the number of S mutations. By contrast, for the experimental data, the number of S and NS mutations is equivalent. Assuming that there is little or no selection pressure on S mutations, we can extrapolate the number of NS mutations before selection. So, in the TEV 2015 experiment [13], the total number of mutations before selection would be 300 mutations distributed in 75 S and 225 NS mutations, 278 mutations for the PVY 2015 experiment [11], and 1077 mutations for the PVY 2017 experiment [12]. We can see that for two datasets, before selection, the minimum required of 1000 mutations previously defined to obtain a robust analysis was not reached. Although not statistically significant, these two datasets reflect true natural selection processes from a larger set of mutations on different pathosystems and are presented as such. 

The amount of S is rigorously proportional with the protein length irrespective of its function. This is not the case for NS, and some proteins contain more mutations than others. For instance, it appears that P1 significantly accumulates more NS than HC-Pro, P3, CI, NIb, and CP (*p* < 0.02; Z test) (Figure 2 and Appendix A). As a matter of fact, P1 proteins contain the highest levels of ID in potyviral proteomes [9]. To further investigate the potential link between ID and NS mutation accumulation, the analysis was no longer conducted on individual proteins but on all IDRs and ORs distributed along the coding sequences in the viral genomes.

### 3.3. Distribution of NS and S Mutations in IDRs and ORs

In order to analyze the distribution of each type of mutation in the IDRs or ORs, we defined the ratio *R* for synonymous mutations as:(1)$$R_{s}=\frac{{\%\mathrm{IDR}}_{S\;}}{{\%\mathrm{OR}}_{S\;}\;}$$
with ${\%\mathrm{IDR}}_{S}\;\mathrm{and}\;{\%\mathrm{OR}}_{S}$ defined as
(2)$${\%\mathrm{IDR}}_{S}=\frac{Number\;of\;S\;mutations\;in\;IDR}{Total\;number\;mutations\;in\;IDR}*100\;$$
(3)$${\%\mathrm{OR}}_{S}=\frac{Number\;of\;S\;mutations\;in\;OR}{Total\;number\;mutations\;in\;OR}*100\;$$

Equations (1)–(3) also apply for the calculation of *R*_NS_, the ratio R representing non-synonymous mutations.

The ratio of synonymous mutations between IDRs and ORs deduced from the experimental data were close to 1 (Figure 3 and Appendix A). This ratio is comparable to that obtained by simulation which mimics random mutations and reflects the absence of impact of synonymous mutations at the protein level.

By contrast, for NS mutations, a large and significant difference between the experimental and simulated data could be observed (*p* < 0.02, χ^2^ for each of the four simulations, Table 2). Indeed, the ratio higher than 1 observed in the case of the experimental data indicates an over-representation of NS mutations within the IDRs compare to the ORs (Figure 3). In the case of PVY, this difference with simulated mutations was only verified for the PVY 2017 dataset [12] (Table 2).

With respect to the analysis of natural diversity, no differences were observed between S mutations within IDRs and ORs, in agreement with simulated data. By contrast, a significant over-representation of NS mutations was observed in the IDRs for all viruses (Figure 3 and Appendix A).

Altogether, those results indicate that IDRs are more prone to accumulate NS mutations than more structured regions at both shorter (experimental evolution) and longer (natural diversity) evolutionary time-scales.

### 3.4. Comparison of the Physicochemical Disturbance of Amino Acid Substitutions in Potyviral Intrinsically Disordered versus Ordered Regions

We also analyzed possible amino acid substitution biases between IDRs and ORs with respect to their physico-chemical properties. To compare the physico-chemical nature of the substitutive amino acids (NS mutations) in IDRs and ORs, we used the BLOSUM62 matrix [20]. This matrix uses the natural diversity between very conserved regions of evolutionarily divergent protein sequences. The set of sequences is aligned to a reference sequence. For each position where a substitution occurs, the probability of occurrence of each of the 19 other amino acids is calculated, resulting in score values ranging between −4 and 11. The higher the score, the higher the likelihood of substitution. It was observed that the highest replacement probabilities are correlated to amino acids with similar physico-chemical properties (e.g., charges, hydrophilicity-hydrophobicity, amino acid size) [21]. Upon amino acid substitutions, the more drastic physicochemical changes are (the lower the score value), the more destabilizing these changes are in terms of structure. 

For each virus (TEV, PVY, and TuMV), we assessed whether the natural selection discriminated differently within IDRs and ORs for amino acid substitutions with respect to their impact on biophysical changes. For each type of region (IDRs or ORs), a comparative statistical analysis (Dunn test) was thus performed between the natural diversity, experimental, and simulated datasets (Table 3 and Appendix A).

For PVY and TuMV, the BLOSUM62 scores associated to amino acid substitutions in IDRs of biological and simulated data belong to the same statistical group. In the case of TEV, the natural diversity data shows a slight difference for three of the four simulations. For TEV, this diversity was represented by a set of only 6 genomes, while those of PVY and TuMV were illustrated by 100 genomes each. This could explain the apparent discrepancy. In contrast, for both the natural diversity and experimental evolution of all three potyviruses, amino acid substitutions present in the ORs have a significantly higher BLOSUM62 score than the simulated data. The high BLOSUM62 score observed as associated to ordered regions supports the idea that amino acids substitutions occurring in those regions are globally poorly destabilizing at the physicochemical and structural level. Reciprocally, with lower BLOSUM62 scores than the ones observed in ORs, IDRs would be more permissive to drastic physicochemical changes. Hence, the substitutions in the ORs are very constrained by the conservation of the physico-chemical properties of the amino acids. This feature is not found in the IDRs where the substitutions tend to be more random.

### 3.5. Are NS Mutations in IDRs Driven toward the Conservation of Disorder Promoting Amino Acids

We investigated whether the conservation of disorder during evolution could be a selection criterion using amino acid disorder promoting scores (see material and method section) which were attributed to each of the 20 amino acids [18,22]. Amino acid residues were grouped into order promoting, neutral, or disorder promoting based on their scores, ranging from 0 (amino acid most frequently present in ORs) to 1 (amino acid most frequently present in IDRs).

We first examined whether substitutions were preferentially targeting order or disorder promoting amino acids in both IDRs and ORs. We did not observe any significant differences between biological data and simulated data within either IDRs or ORs (Appendix A). We concluded that there is no natural tendency for evolution to target substitutions preferentially toward order or disorder promoting amino acids.

Then, we considered the possibility that non-synonymous mutations (NS) could preferably give order or disorder promoting amino acids (Appendix A). We observed unbiased random substitution. This, both in ORs or IDRs, was in accordance with simulations. Finally, we aimed at assessing a possible tendency for substitution by amino acids that are more prone to promote order or disorder (Appendix A). At each position where a NS mutation was observed, we calculated the difference in disorder-promoting score between the amino acid in the reference genome and the replacing amino acid in each of the genomes describing the diversity in the biological data. Again, we did not observe significant differences between naturally selected and simulated mutations. We did not detect any differences between biological and simulated data in the disorder-promoting score for the non-synonymous mutations (either in the IDRs or the ORs). However, global disorder is generally conserved during evolution [23,24,25], and more specifically in RNA viruses [5,9]. Therefore, the analysis of local substitutions does not reflect this evolutionary trend that can be observed globally at the scale of a protein region. It turns out that the analysis of substitutions in terms of physico-chemical modulations seems more relevant than the use of the disorder-promoting score.

### 3.6. Comparative Assessment of Codon Volatility between IDRs and ORs

We examined the hypothesis that intrinsic disorder could be selected to generate a pool of mutations available for adaptive function. To obtain the amino acid diversity observed in IDRs, two successive processes, namely the generation of non-synonymous mutations and their selection, are involved. The first process can be favored by codon volatility, the probability that a random point mutation in a codon is non-synonymous. Practically, it is calculated as the proportion of a codon’s point mutation neighbors that code for different amino acids [19]. We investigated whether the nucleotide sequences encoding IDRs used codons of higher volatility than the sequences of ORs, thus favoring the generation of non-synonymous mutations. In this study, the 6 TEV accessions used display more than 3000 mutations. Volatilities within the IDRs and ORs of these six genomes were compared. The same volatility was observed in all IRs and ORs, revealing an homogeneity within the population of genomes. We then compared the volatility of IDRs and ORs within each genome; however, no difference was observed. Consequently, for the two other species, IDR vs. OR volatility was only compared within the genome of reference. IDRs and ORs display the same volatility (Appendix A) (U test; *p* value > 0.01). However, one could ask how far this natural potyviral volatility differ from a randomly generated volatility. For each of the TuMV, PVY, and TEV reference genomes, we randomly produced 1000 alternative nucleotide sequences, encoding the same amino acid sequences with single nucleotide synonymous mutation in each codon. For each viral species, the mean volatility within ORs and IDRS of these random synonymous variants were compared to the mean volatility found in the natural genomes. As expected, a significant statistical difference was observed between the simulated and natural volatilities of the three genomes in both IDRs and ORs (Appendix A) (U test; *p* value < 0.01), which likely results from potyviral codon usage.

## 4. Discussion

Potyviruses constitute, together with begomoviruses, the two largest viral genera described to date among plant viruses [26,27]. Potyviruses are very damaging to field crops and embrace a very wide host range [28]. Most of them are generalists and, as such, provide a rich model for studying viral adaptation. In this study, we tested the hypothesis that among these viruses, the amount of non-synonymous mutations (NS) was greater in the disordered regions of their proteomes than in the ordered regions. We analyzed the distribution of mutations in the genomes of potyviruses belonging to three different species (i.e., PVY, TEV and TuMV) which are representatives of their genus. The datasets used included viral genomes resulting from both shorter (experimental evolution) and longer evolutionary scales along with the use of natural diversity. An analysis of the two datasets showed that IDRs and ORs are subject to different evolutionary mechanisms, with disordered regions evolving towards significantly more amino acid polymorphism than ordered regions. The selection pressure that applies to ordered regions thus tends towards a conservative evolution while that which applies to disordered regions rather supports a divergent evolution. It is quite easy to understand the evolutionary mechanism at work in the ordered regions. These protein regions have a strong structure–function relationship. At the molecular level, these regions are defined by geometries of constrained atomic interactions with few degrees of freedom and well-packed hydrophobic cores. These regions have significantly higher BLOSUM62 scores than would result from random substitutions. Substituted amino acids have physicochemical natures close to those of the original amino acids. Conversely, the lower topological requirement in disordered regions results in substitutions close to the random substitution pattern. Within ordered regions, mutations compensate for each other in order to prevent instability according to an epistatic model. On the long term, such compensation leads to changes in sequence and function (protein evolvability) [7,29]. In these regions, the selection pressure strongly operates to preserve functions which result in amino acids conservation. In disordered regions, the notion of structural stability is less relevant and amino acids substitutions may have less functional impact [5]. This could constitute an alternative model for protein evolvability, presumably on a shorter evolutive timeline consistent with the rapid adaptation characteristic of viruses. From an evolutionary standpoint, the dogma of the structure-function relationship (conservation of function requiring conserved structures and therefore close substitutions) requires tempering in the case of IDRs.

We did not observe greater codon volatility in the disordered regions than in the ORs, which could explain the greater amino acid polymorphism observed in the disordered regions. Thus, with respect to the volatility criterion, we have no evidence to support that amino acid polymorphism in disordered regions undergoes positive selection, generating a potential adaptive pool for the virus. Although amino acid polymorphism in these regions may participate in potyvirus adaptation, the conservation of intrinsic disorder during evolution mainly results from a selection pressure dictated by the essential biochemical functions associated to virus replication in the host. In any case, the mutational robustness and diversity that arise from the selection of structure-function relationships within IDRs is likely to favor the adaptive potential of the virus. However, the hypothesis that IDRs could also be conserved in the course of evolution as reservoirs for an adaptive potential remains difficult to assess.

It should be expected that the synonymous codon usage pattern of viruses would be shaped by selecting specific codon subsets to match the most abundant host transfer RNAs (tRNAs). However, the codon usage of many viruses is very different from the optimal codons present in the host [30]. Interestingly, it was recently reported that codon usage in virus IDRs is less optimized for the host than in ORs [31]. In the case of NS mutations, this is in line with our observation that IDRs are more robust to mutations than OR (and thus evolve faster). This prevents fixation of codons optimized for the host (Figure 4). The preservation of codon diversity in these regions may also provide a reservoir for a faster adaptation of the viruses to various hosts.

Due to the presence of less frequent codons in IDRs, the corresponding pools of loaded tRNAs in the host cell are lower than those of abundant codons. Consequently, translational dynamics is likely to be slowed down when the ribosome machinery enters a mRNA sequence encoding for disordered regions [32,33]. This may result in an instability of the translation product [34]. IDRs are generally taken over either co-translationally or post-translationally by chaperones. This handling does not favor the selection of optimized codons and contributes to the preservation of amino acid polymorphism in IDRs. There is an intricate interplay of molecular chaperones and protein disorder in the evolvability of protein networks [35].

Taken all together, the data obtained unambiguously show that potyvirus IDRs and ODRs follow very different evolutive paths with respect to their mutational robustness. These results forced the authors to consider the hypothesis that during selection, adaptive solutions could emerge from the amino acid polymorphism carried by IDRs.

The hypothesis that IDR mutational robustness could provide a reservoir of mutations for the adaptive potential of viruses is not very well documented. The first experimental evidence that protein intrinsic disorder could favor host resistance breaking was reported in [10]. More recently, an in silico study on SARS-CoV-2 virus revealed that, in the course of time, mutations appearing in the N-protein central IDR were associated with an enhancement of its disorder level. The authors suggested that the higher flexibility of this region could enhance the β-interferon antagonist function of this protein [36]. Ultimately, establishing general link between intrinsic disorder and RNA virus adaptation could promote this structural feature as a new criterion to include in the design of antiviral control methods. Indeed, owing to their adaptive abilities, viral IDRs could constitute poor targets for drugs, vaccines, and host genetic resistance which constitute the main antiviral strategies in animal and plants, respectively.

## Figures and Tables

**Figure 1 viruses-14-01959-f001:**
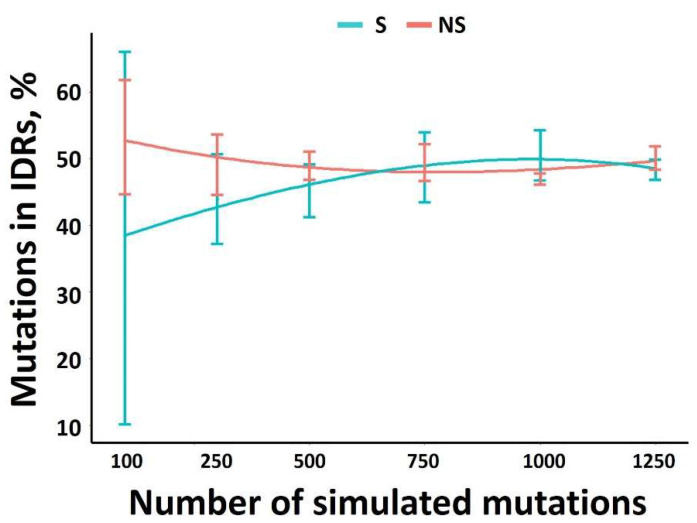
The % of S and NS mutations in IDRs versus the mutations number in the TEV genome. For a given number of mutations, four independent simulations were run.

**Figure 2 viruses-14-01959-f002:**
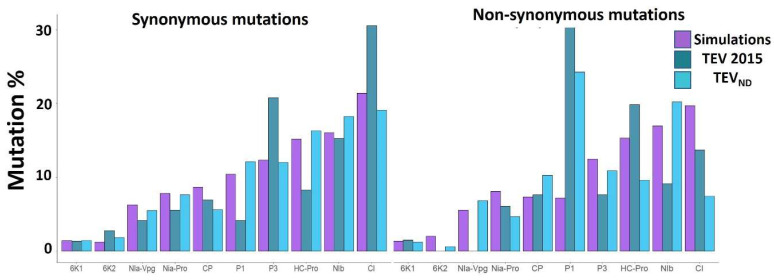
Mutation % in the TEV proteins from the experimental evolution [13], natural diversity and simulations. For PVY and TuMV, please refer to the Appendix A. The proteins are sorted from the smallest to the largest, left to right: 6K1, 6K2, Nia-VPg, Nia-Pro, CP, P1, P3, Hc-Pro, Nib, and CI. % mutation = (number of NS or S mutation within each protein)/(total number of NS or S mutations within the whole genome).

**Figure 3 viruses-14-01959-f003:**
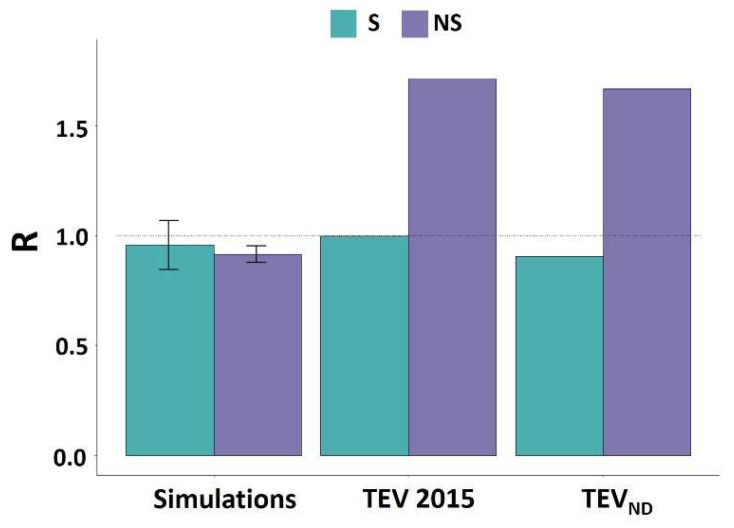
Ratio between the percentage of mutation (S or NS) present in IDRs and ORs for the TEV genome. For datasets from the two other studies [11,12] regarding TuMV and PVY, please refer to the Appendix A.

**Figure 4 viruses-14-01959-f004:**
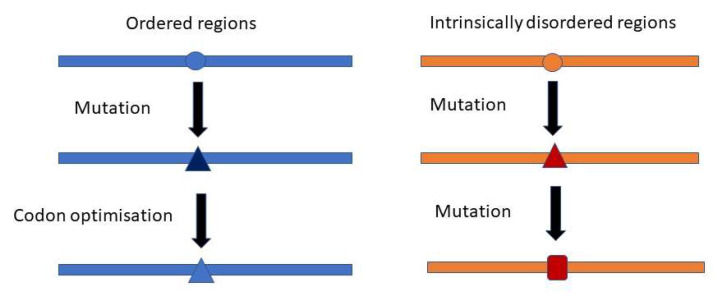
Mutational robustness of IDRs and codon optimization. The low rate of non-synonymous mutations in the ORs allows the optimization of the sequence towards the selection of abundant codons in the host. The high rate of non-synonymous mutations in IDRs prevents this optimization.

**Table 1 viruses-14-01959-t001:** Correlation coefficient (R²) between coding sequence length of the TEV proteins and the mutations (S or NS). Experimental evolution: TEV 2015 [13], PVY 2015 [11], and PVY 2017 [12]. TEV_ND_, PVY_ND_, and TuMV_ND;_ ND, natural diversity. Simulations: four in silico replicates.

	**S**	**NS**
TEV 2015	0.63	0.19
TEV_ND_	0.94	0.16
Simulations	1	0.98
	**S**	**NS**
PVY 2015	0.93	0.12
PVY 2017	0.78	0.01
PVY_ND_	0.96	0.35
Simulations	0.96	0.97
	**S**	**NS**
TuMV_ND_	0.95	0.09
Simulations	0.92	0.98

**Table 2 viruses-14-01959-t002:** *p* values of Xhi^2^ test for percentage of NS mutations in IDRs between simulated and experimental data (TEV 2015, PVY 2015, PVY 2017) or natural diversity (TEV_ND_, PVY_ND_, TuMV_ND_). There was no experimental datasets available for TuMV. Significance, *p* < 0.05.

Simulations	TEV 2015	TEV_ND_	PVY 2015	PVY 2017	PVY_ND_	TuMV_ND_
**A**	0.014	2.10 × 10^−6^	0.52	0.04	0.0005	3.10 × 10^−8^
**B**	0.026	1.10 × 10^−5^	0.46	0.03	0.0001	5.10 × 10^−8^
**C**	0.018	5.10 × 10^−6^	0.33	0.01	8.10-6	2.10 × 10^−8^
**D**	0.028	2.10 × 10^−5^	0.27	0.0049	1.10-7	6.10 × 10^−9^

**Table 3 viruses-14-01959-t003:** Differences in physicochemical properties associated with amino acid substitutions were assessed using scores derived from the BLOSUM62 substitution matrix. For each type of region (IDRs or ORs), virus biological and simulated data were distributed into statistical groups (a, b, and c) by running a Dunn test (*p* value adjustment method: Bonferroni) for the (A) PVY genome, (B) TEV genome, and (C) TuMV genome.

A	IDRs	ORs	B	IDRs	ORs	C	IDRs	ORs
PVY 2015	abc	ab	TEV 2015	ab	ab	TuMV_ND_	a	a
PVY 2017	abc	a	TEV_ND_	b	b	Sim A	a	b
PVY_ND_	ac	a	Sim A	a	a	Sim B	a	b
Sim A	bc	c	Sim B	a	a	Sim C	a	b
Sim B	bc	c	Sim C	a	a	Sim D	a	b
Sim C	b	bc	Sim D	ab	a			
Sim D	bc	c

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
