# Peer review of "Analysis of the Contribution of Intrinsic Disorder in Shaping Potyvirus Genetic Diversity"

_viruses, 2022, doi:10.3390/v14091959_

Round 1

Reviewer 1 Report

General comments

 The central question raised by this manuscript entitled “Analysis of the contribution of intrinsic disorder in shaping potyvirus genetic diversity” is whether regions predicted as intrinsically disordered (IDRs) in potyviral proteomes are more likely to accumulate amino acid substitutions (and the corresponding nucleotide sequences more likely to accumulate non-synonymous NS mutations) than more ordered regions (ORs), thus constituting a reservoir of amino acid polymorphism.

To address this question, the authors exploit data from three previously published high-throughput sequencing experiments for two potyvirus species (TEV and PVY), two of which are experimental evolution assays involving sequential passages on different hosts. Additional data used in the study are publicly available complete genome sequences from potyvirus natural diversity for three potyvirus species (TEV, PVY and TuMV). The aim is to study both short-term (mutations accumulated in in planta viral populations) and long-term evolution scales (natural diversity) .

In order to perform this comparison between IDRs and ORs, the distribution of mutations in the biological datasets is compared to that of simulated datasets of randomly generated mutations in the viral genomes. This is the main strength of the manuscript, because the use of simulated datasets allows the comparison of IDRs and ORs in an unbiased manner, not used in previous analyses.

Although this review is well written and thoroughly referenced, one of my main concerns is that the authors do not describe in detail the experiments from which they retrieved the HTS data and do not explain precisely what part of the data they are using and how.

The main characteristics of the experimental data should be listed (plant host(s), nature of initial inoculum, number of passages…) for each of the 3 datasets (perhaps this could be summarized in a table) so that the reader does not need to constantly refer to the original publications to gather this information.

My guess is that the authors used all the mutations detected at some point in the experiments indistinctly, regardless of the host plant (natural host or “new” host, highly susceptible or partially resistant host…) , the passage number, and the frequency of these mutations, in order to have the largest possible dataset of “experimental” mutations in the 2 potyviral genomes. However, if this is the case, it is difficult to make conclusions on the potentially adaptive nature of the mutations. The only conclusions that can be made is on the pattern of accumulation of non-synonymous mutations in IDRs and ORs to see whether they are distinct from what would be expected by chance.

Moreover the authors have estimated the number of mutations that need to be considered in order to obtain an accurate estimation of the distribution of mutations in the viral genomes (lines 142-167, and 188-196). Based on these criteria, only the “PVY 2017” experiment and the natural diversity datasets qualify. Therefore the question that comes to mind is whether it makes sense to analyze the two other experimental datasets (“TEV 2015” and “PVY 2015”).

Another of my concerns is the fact that the authors have not defined the terms and concepts that are central to the manuscript (like “mutational robustness” - there are 9 occurrences of “mutational robustness” in the manuscript but the term is never defined - “evolvability”, “adaptive mutation”) and that, consequently, the reader is left in some doubt as to the precise objectives and conclusions that are being drawn.

In reference 5 (Walter et al. 2019) to which the 3 co-authors contributed, mutational robustness is defined as “the ability of an organism to maintain its phenotype despite the disturbances caused by mutation accumulation”. The results presented in this manuscript do not directly address mutational robustness defined in this way. Indeed there are no functional analyses of the role of the different mutations considered in this study to address the function of the viral proteins and/or their contribution to viral fitness in different hosts/environments. Therefore no conclusions can be made concerning the adaptive nature of the mutations included in the 3 datasets or the stability of the viral phenotypes in the face of mutations accumulated in IDRs versus ORs.

Some of the results presented in the manuscript (those presented first, lines 168-200, concerning the correlation between the number of mutations and protein size) do not appear to answer the central question of this paper, or the authors have not shown how they may contribute to address this question. Only results that are relevant and that help answer the questions being addressed should be presented.

The first subsection of the Discussion section discusses results presented in the manuscript, however the following sections do not. Codon volatility and codon usage in IDRs versus ORs are discussed, although no results concerning these two parameters are presented. Was a subsection dedicated to codon volatility and usage in IDRs versus ODs removed from a previous version of the Results section but conserved in the Discussion section?

The discussion should compare the results obtained to what is known for other viruses or organisms. The perspectives of the work presented (e.g. functional analyses?) could also be proposed. Finally, what are the implications of the results? Notably, are there any practical consequences, regarding the management of viral diseases in plants?  

Specific comments

Line 13 Please define mutational robustness : is it the "mutational permissiveness" stated above?

Definitions I found in the literature:

"Mutational robustness: is the extent to which the phenotype of an organism (i.e., morphology or functional performance) remains constant in spite of mutations to its genotype."

(https://doi.org/10.1016/j.tig.2015.04.008)

"de Visser and colleagues have provided a useful working definition of robustness: “Robustness is the invariance of phenotypes in the face of perturbation” (REF. 13). They further classify robustness on the basis of the source of perturbation, which can be either genetic or environmental." (Lauring, A., Frydman, J. & Andino, R. The role of mutational robustness in RNA virus evolution. Nat Rev Microbiol 11, 327–336 (2013). https://doi.org/10.1038/nrmicro3003)

Line 14 The DOI should not be included in the abstract

Line 15used to distinguish a possible selection phenomenon in the biological data sets from randomly generated mutations”

Why only a selection phenomenon? The assumption that mutation alone vs. mutation + selection will be compared by comparing simulated versus experimental data is incorrect or at least oversimplified. Other processes aside from selection, like genetic drift or complementation in the viral population at a given time between different variants can shape the virus mutational landscape.

Line 16 “several short-term experimental evolution datasets”

Not all datasets correspond (strictly speaking) to experimental evolution experiments (cf. Kutnjak et al. 2015).

Concerning the datasets : were all the data used? Or only part of the data (for example mutations from the last passage on the “new host” compared to the initial strain and not data on the natural host?)

Lines 20-21 Moreover, the substitutions in the ORs are very constrained by the conservation of the physico-chemical properties of the amino acids. This feature is not found in the IDRs where the substitutions tend to be more random”

Here (in the abstract this conclusion is well stated) but not in the results section (lines 260-262).

Line 24 “during selection, adaptive solutions could emerge from the amino acid polymorphism carried by IDRs”

Here the message is clear: mutations accumulated in IDRs can be selected and offer adaptative solutions for viruses. This is not something surprising and references cited in the manuscript already pointed to this (including Charron et al. 2016 from the same group showing the cooccurrence between conserved IDRs in certain potyviral proteins and amino-acid polymorphism).

Lines 39-40 “inter-residue contact density” should be defined precisely

Lines 35-45 This part of the  introduction is basically a cut and paste of the introduction in Walter et al. 2019 (reference 5) to which the 3 coauthors contributed.

Line 77 The short-term scale data analyzed consisted in high-throughput sequencing (HTS) retrieved from three independent evolution experiments, i.e. PVY [10,11], and TEV [12].”

Strictly speaking, not all are evolution experiments.

Lines 83-84 “To evaluate the impact of disordered versus ordered region on potyvirus evolvability (i.e. mutational robustness) at a higher scale of evolution, genomic sequences from TuMV,TEV and PVY natural diversity were also retrieved”

This gives the impression that evolvabilty and mutational robustness are synonymous whereas they are not, and a clear definition of each would be useful. "A key unresolved question, then, is whether mutational robustness increases or decreases evolvability, which we refer to as the capacity of a virus to increase in fitness through adaptation" (Lauring et al. 2013, doi: 10.1038/nrmicro3003)

Line 87  “potyvirus genomes were artificially mutated in silico according the viral replicase features, to mimic the genetic diversity obtained in the absence of selection and, among others, effects of protein structural determinants”

What genome sequence(s) were the input, especially in the case where experimental evolution was initiated with an inoculum not originating from a cloned virus?

In the natural diversity datasets, was a single initial sequence included, and was it the same in the different simulations? How was this sequence chosen?

Line 101 The study of Cuevas et al 2015 (TEV 2015), [12] evolved the TEV on two different host, Nicotiana tabacum and Capsicum annuum”

Does this mean data on both hosts were analyzed together?

Wouldn’t it make more sense to analyze them separately?

Lines 102-103 Table S1 compiles all the resulting mutations of experimental datasets.”

Do these include mutations detected at all passages when the data come from experimental evolution, and on all hosts (for example natural and "new" host for TEV)?

Moreover, since the data come from HTS experiments, how are the different frequencies of the mutations taken into account (major versus minor - often barely above 1% -mutations or variants)?

In other terms, are the results presented based on all of the SNPs identified regardless of their frequencies, of the host, and of the passage number?

It would be interesting to test whether there is a correlation between the frequency of variants and their presence in disordered regions (which might reflect the fitness of the variants, although processes like genetic drift might favor deleterious mutations).

Line 110 The distribution of mutations in the virus sequence is the sum of the contribution of viral polymerase errors and of the subsequent selection according to structure-function relationships.”

This statement is an oversimplification. What about bottlenecks and the associated genetic drift and possible complementation between variants present at a given time, or compensatory mutations that can help maintain an otherwise deleterious mutation in a given haplotype? The function of chaperones to buffer the deleterious effect of certain mutations on protein stability as detailed in lines 357-365 of the discussion section should also be considered.

Line 120 “We generated n variants from the original potyvirus sequence, with each variant bearing one SNP

In each case (PVY 2015, PVY 2017, TEV 2015 and TEVND, PVYND, TuMVND) what is used as "the original potyvirus sequence" (cf. comment concerning line 87)?

Please indicate the different values of “n” that were tested.

Line 130 Promotor scores, ranging from 0 to 1”

Promoter/promotor/disorder-promoting : depending on the figure, table or part of main text different spellings and terms are used. Choosing the term « disorder-promoting [score] »  as in lines 140, 264, 266, 275, and 277 and sticking with it throughout the manuscript and figures would make things clearer.

Line 158 “monitoring the evolution of the R² coefficient as a function of the mutation number.”

The R2 coefficient is not defined here.

It is in the caption of Fig S1: “R², the coefficient referring to the correlation between percentage of mutations (S or NS) and protein length in the TEV genome”

Moreover, I am not sure I understand why this R2 coefficient is considered here: to make sure the number if mutations is correlated to protein length? Is this important in the current study?

Line 168 “Correlation assessment between protein length and number of mutations”

This should come before the previous subsection, although, again, I don't see how these results are useful here. Is the point to show that more NS mutations accumulate in P1 and that P1 has more ID regions or simply that NS mutations  "are thus not accumulated homogeneously along the potyvirus proteome."?

Does the heterogeneity correlate with IDRs? In other words do NS mutation hotspots coincide with IDRs?

Lines 169-170 “propensity of each potyvirus proteins to accumulate adaptive non-synonymous (NS) versus synonymous (S) mutations »

Please define « adaptive » because it is not clear whether “adaptive” is used as a synonym of “non-synonymous” or not. It might be useful to point out that mutations that are deleterious in the current environment and current plant host may be adaptive under different conditions.

Lines 170-172 The number of NS or S mutations observed in each protein coding sequence divided by the total protein length”

Data not shown here, only the R2 between the number of mutations and protein length is presented in Table I.

It would be preferable to illustrate this result with a plot comparing the number of mutations to protein sequence length.

Again, this result is of interest in the present study only if it is interpreted in the light of the presence/absence of IDRs in each protein, otherwise, in my opinion, this result is not clearly relevant to the aim of this paper (if the aim is to measure whether IDRs accumulate more NS mutations than what is expected at random).

Lines 179-182 Non-synonymous adaptive mutations, which reflect the viral amino acid polymorphism, are thus not accumulated homogenously along the potyvirus proteome.”

As in lines 169-170 it is seems as if the non-synonymous nature of a mutation implies that it is adaptive.

Using the term “potentially adaptive” would be safer.

Lines 194 sqq and 1077 mutations for the PVY 2017 experiment [11]. We can see that for two datasets, before selection, we do not reach the minimum required of 1000 mutations previously defined to obtain a robust analysis.”

According to the authors' criteria the PVY 2017 study is the only statistically tractable data set. So does it make sense to analyze data from the two other datasets?

Lines 199 sqq For instance, it appears that P1 significantly accumulates more NS than HC-Pro, P3, CI, NIb and CP (P < 0.02; Z test) (Figure 2 and Figure S2).”

How does this relate with IDR content in the different viral proteins?  It might be useful to state that P1 is one of the viral proteins that have the most IDRs.

However, are these results new? Was it not previously described that IDR-rich potyviral proteins accumulate more NS mutations than the other viral proteins?

Figure 2 Mutation % in the TEV proteins from the experimental evolution [12], natural diversity and simulations »

Please define: what exactly is the “mutation %”?

% of all mutations in the genome that occur in the given protein-coding sequence?

Line 229 Altogether, those results indicate that IDRs are more prone to accumulate adaptive mutations than more structured regions, at both short (experimental evolution) and longer (natural diversity) evolutionary time-scales.”

Why adaptive? Just NS and potentially adaptive. The authors need to be careful about the use of terms such as “adaptive”.

Lines 237-245 could be moved to the Material and Methods section, otherwise one might wonder whether this matrix was obtained as part of this study based on the specific datasets presented here.

Lines 249 sqq

It would be interesting to not only have the results of the Dunn test but also visualize Blosum62 scores along the genome/proteome and visualize how these scores are distributed compared to predicted IDRs and ORs. See figure 5 from Kutnjak et al. 2015, for example (circular representation of PVY genome with plotted frequencies of SNPs in the sRNA and VP pools, corresponding BLOSUM62 amino acid substitution scores): adding a layer of data specifying the position of predicted IDRs and ORs to a similar representation would be more informative. And the comparative statistical analysis would be less abstract.

Line 260

Why not compare IDRs in experimental versus simulated data rather than compare IDRs versus ORs? As is done in Table(s) 3? Isn't there a bias, as has been explained in the beginning of the manuscript?

Table 3 BLOSUM not BLOSSOM

Lines 284-285 “We did not detect any differences between biological and simulated data in the promoter score for the synonymous mutations, either in the IDRs or ORs.”

I don't understand this: how can there be a change in disorder-promoting score for synonymous mutations (no change in encoded amino acid, therefore no change in disorder-promoting score)?

Line 326 “Does amino acid polymorphism in potyvirus proteome IDRs undergoes positive selection?”

How does this title relate to the content of the subsection which concerns codon volatility?

The link between codon volatility and positive selection (suggested by this title) isn’t obvious to me and deserves an explanation. Ref 28 (doi: 10.1534/genetics.104.034884) cited in the manuscript concludes that "codon volatility has only limited utility for detecting positive selection at the DNA sequence level"

Lines 327 sqq: codon volatility

Was a subsection dedicated to codon volatility in IDRs versus ODs removed from the Results section but conserved in the Discussion section?

Line 330

How can mutation be favored by codon volatility ?

Lines 347-355 codon usage

What results presented in this manuscript relate to this discussion?

Lines 357-365 codon usage and chaperones

Where does this result come from? Was an analysis of codon usage performed on the potyvirus sequences? Was the codon usage/composition compared in potyviral IDRs and ORs? What are the results? Otherwise this aspect is discussed in the manuscript but there are no corresponding results.

Author Response

1
Lafforgue et al. Analysis of the contribution of intrinsic disorder in shaping potyvirus genetic diversity

Answers to reviewer1. Viruses, MS#1835658

We thank reviewer 1 (R1) for his/her in depth analysis of this work. These remarks were quite useful
in our attempt to improve the manuscript and open perspectives for our oncoming work. Point to point
answers have been inserted in bold within R1 text.

Note that line number correspond to the new version of the manuscript once mods have been
accepted.

Reviewer 1

General comments

The central question raised by this manuscript entitled “Analysis of the contribution of intrinsic
disorder in shaping potyvirus genetic diversity” is whether regions predicted as intrinsically disordered
(IDRs) in potyviral proteomes are more likely to accumulate amino acid substitutions (and the
corresponding nucleotide sequences more likely to accumulate non-synonymous NS mutations) than
more ordered regions (ORs), thus constituting a reservoir of amino acid polymorphism.

To address this question, the authors exploit data from three previously published high-throughput
sequencing experiments for two potyvirus species (TEV and PVY), two of which are experimental
evolution assays involving sequential passages on different hosts. Additional data used in the study are
publicly available complete genome sequences from potyvirus natural diversity for three potyvirus
species (TEV, PVY and TuMV). The aim is to study both short-term (mutations accumulated in in planta
viral populations) and long-term evolution scales (natural diversity) .

In order to perform this comparison between IDRs and ORs, the distribution of mutations in the
biological datasets is compared to that of simulated datasets of randomly generated mutations in the
viral genomes. This is the main strength of the manuscript, because the use of simulated datasets
allows the comparison of IDRs and ORs in an unbiased manner, not used in previous analyses.

Although this review is well written and thoroughly referenced, one of my main concerns is that the
authors do not describe in detail the experiments from which they retrieved the HTS data and do not
explain precisely what part of the data they are using and how.

The main characteristics of the experimental data should be listed (plant host(s), nature of initial
inoculum, number of passages...) for each of the 3 datasets (perhaps this could be summarized in a
table) so that the reader does not need to constantly refer to the original publications to gather this
information.

My guess is that the authors used all the mutations detected at some point in the experiments
indistinctly, regardless of the host plant (natural host or “new” host, highly susceptible or partially
resistant host...) , the passage number, and the frequency of these mutations, in order to have the
largest possible dataset of “experimental” mutations in the 2 potyviral genomes. However, if this is the
case, it is difficult to make conclusions on the potentially adaptive nature of the mutations. The only
conclusions that can be made is on the pattern of accumulation of non-synonymous mutations in IDRs
and ORs to see whether they are distinct from what would be expected by chance.

2
Moreover the authors have estimated the number of mutations that need to be considered in order
to obtain an accurate estimation of the distribution of mutations in the viral genomes (lines 142-167,
and 188-196). Based on these criteria, only the “PVY 2017” experiment and the natural diversity
datasets qualify. Therefore, the question that comes to mind is whether it makes sense to analyze the
two other experimental datasets (“TEV 2015” and “PVY 2015”).

The number of mutations in TEV 2015 and PVY 2015 is too low to be analyzed significantly. However,
the selection process started from a higher number of mutations. Hence, we stated in the text (lines
204-206), our choice to maintain these data in the manuscript.

Another of my concerns is the fact that the authors have not defined the terms and concepts that are
central to the manuscript (like “mutational robustness - there are 9 occurrences of “mutational
robustness” in the manuscript but the term is never defined - evolvability”, “adaptive mutation”) and
that, consequently, the reader is left in some doubt as to the precise objectives and conclusions that
are being drawn.

In reference 5 (Walter et al. 2019) to which the 3 co-authors contributed, mutational robustness is
defined as “the ability of an organism to maintain its phenotype despite the disturbances caused by
mutation accumulation”. The results presented in this manuscript do not directly address mutational
robustness defined in this way. Indeed there are no functional analyses of the role of the different
mutations considered in this study to address the function of the viral proteins and/or their
contribution to viral fitness in different hosts/environments. Therefore no conclusions can be made
concerning the adaptive nature of the mutations included in the 3 datasets or the stability of the viral
phenotypes in the face of mutations accumulated in IDRs versus ORs.

Some of the results presented in the manuscript (those presented first, lines 168-200, concerning the
correlation between the number of mutations and protein size) do not appear to answer the central
question of this paper, or the authors have not shown how they may contribute to address this
question. Only results that are relevant and that help answer the questions being addressed should be
presented.

The first subsection of the Discussion section discusses results presented in the manuscript, however
the following sections do not. Codon volatility and codon usage in IDRs versus ORs are discussed,
although no results concerning these two parameters are presented. Was a subsection dedicated to
codon volatility and usage in IDRs versus ODs removed from a previous version of the Results section
but conserved in the Discussion section?

In this study, the six TEV accessions used display more than 3000 mutations. Volatilities within the
IDRs of these six genomes were compared. The same was done for the ORs. The same volatility was
observed in all IRs and ORs, revealing an homogeneity within the population of genomes. We then
compared the volatility of IDRs and ORs within each genome. No difference was observed.
Consequently, for the two other species, IDRs vs ORs volatility was only compared within the
genome of reference. Volatility results is now presented at the end of the results section (line 305-
317).

The discussion should compare the results obtained to what is known for other viruses or organisms.
The perspectives of the work presented (e.g. functional analyses?) could also be proposed. Finally,

Reviewer 2 Report

This is a really intresting and valuable manuscript.

L 26 Keywords are missing please complete

L 145 Before the sentence please remove 1+ space

Figure 1. Second row, please remove 1+ space before the sentence

L 199-200 for example the amino- and carboxy-terminal part of potyviral CP is very variable than the core region, because for particle formation this region should be conserved, the protein can’t tolerate the mutation, so it may mention, even there are some references the impact of structure-function of the proteins in the manuscript.

Author Response

Lafforgue et al. Analysis of the contribution of intrinsic disorder
in shaping potyvirus genetic diversity Answers to reviewers. Viruses, MS#1835658

Reviewer 2

This is a really interesting and valuable manuscript.

We thank reviewer 2.

L 26 Keywords are missing please complete

Keywords added

L 145 Before the sentence please remove 1+ space

Corrected

Figure 1. Second row, please remove 1+ space before the sentence

Corrected

L 199-200 for example the amino- and carboxy-terminal part of potyviral CP is very variable than the
core region, because for particle formation this region should be conserved, the protein can’t tolerate
the mutation, so it may mention, even there are some references the impact of structure-function of
the proteins in the manuscript.

Corrected

Round 2

Reviewer 1 Report

General comments

In this revised version, the authors have addressed most of my concerns and the manuscript has been highly improved. A few problems persist and I have annotated the revised manuscript accordingly (see attached pdf file).

Other general comments are listed below.

- The authors have included in this revised version results concerning codon volatility in IDRs and ORs. My main concern now is this codon volatility analysis.

The authors have not detailed the methods they use to calculate codon volatilities of IDRs and ORs, nor have they presented the detailed results. This information should be included in the methods section and results section (e. g. a supplemental table or figure could be included), respectively.

Moreover, I believe codon volatility in IDRs and ORs should not be compared directly, as amino acid composition and the length of these regions constitute a bias, much as they do for the analysis of S and NS mutation distributions in these regions. The authors have been careful to compare these mutation distributions to simulated “control” data, but do not take this precaution when comparing codon volatilities between IDRs and ORs.

It would be preferable to compare codon volatility in IDRs/ORs to a set of simulated synonymous sequences to see whether codon volatility in these regions deviates from what is expected “randomly” given the codon usage in the potyviral genomes.

A method for computing volatility P-values for whole genes has been presented in Plotkin et al. 2004 (doi: 10.1038/nature02458) and further explained in Plotkin et al. 2006 (doi:10.1093/molbev/msl021) and might be applicable to gene regions (IDRs and ORs) of potyviruses. The authors could explore this possibility, although I am not sure whether this is easily applicable to small genomes like potyvirus genomes. This might yield results different from those presented in the manuscript where no difference between codon volatilities in IDRs and ORs was detected.

-  In the beginning of the introduction, mutational robustness is defined as “the capacity to accumulate amino acid substitutions without drastic impairment on protein function”. This definition implies that protein function is unaffected by the accumulated mutations, and therefore based on this definition, strictly speaking, mutational robustness should not be invoked unless protein function is assessed.

Elsewhere in the manuscript no protein function/virus phenotype is assessed but the term “mutational robustness” is used (9 occurrences) in a way that suggests the authors are merely considering the first part of their definition “the capacity to accumulate amino acid substitutions”.

For example in the discussion: “In this study, we tested the hypothesis that among these viruses, the accumulation of non-synonymous mutations (NS) which we simply relate here to mutational robustness was greater in the disordered regions of their proteomes than in the ordered regions.”

The verb “relate to”  is a bit vague. Do the authors mean "define here as" ?

It should be at least stated somewhere in the text that only the first condition (the accumulation of amino acid substitutions) is addressed in this manuscript and that whether the mutations are deleterious or not in certain hosts or environmental conditions remains to be evaluated.

Alternatively, the term mutational robustness should be replaced by another term, or defined differently.

- In the Discussion section, the end of the first subsection “Mutational robustness differences between IDRs and ORs” cited below is confusing to me, particularly now that the previous sentences have been removed.

“Although amino acid polymorphism in these regions may participate in potyvirus adaptation, the conservation of intrinsic disorder during evolution is the result primarily of the second process [what is the first process and where is it mentioned in the previous sentences? Now that the sentence “To obtain the diversity observed in IDRs, two successive processes, namely the generation of mutations and their selection are involved.” has been removed from the discussion, this phrase no longer makes sense], a selection pressure dictated by the essential biochemical functions it [what does “it” refer to here ? “it” should be replaced with “they” (these regions). ] performs to ensure virus replication in the host. In any case, the mutational robustness and diversity that arise from the selection of structure-function relationships within IDRs [does this mean there may be strong structure-function relationships in IDRs after all? Why would they necessarily lead to mutational robustness?] is likely to favor the adaptive potential of the virus. It cannot therefore be excluded that IDRs are also selected according to this last criterion [structure function relationships?], even if this hypothesis remains difficult to assess.”

Is the aim simply to state that, despite high amino acid polymorphism in these regions, IDRs are conserved in potyvirus proteomes and that the explanation for this is that they ensure important biochemical functions in the virus cycle of infection?

- In the Discussion section, the second subsection is dedicated to discussing results from the literature. This is perfectly all right, but it is seems a bit odd to me that this should be made into a subsection. Given the length of the discussion, in my opinion, subsections are not necessary.

Specific comments

See attached pdf file of annotated revised manuscript for specific suggestions and comments.

Author Response

We thank again Rev1 for the fruitful comments and suggestions

Point to point answers are inserted within the text below.

General comments

In this revised version, the authors have addressed most of my concerns and the manuscript has been highly improved. A few problems persist and I have annotated the revised manuscript accordingly (see attached pdf file).

Other general comments are listed below.

- The authors have included in this revised version results concerning codon volatility in IDRs and ORs. My main concern now is this codon volatility analysis.

The authors have not detailed the methods they use to calculate codon volatilities of IDRs and ORs, nor have they presented the detailed results. This information should be included in the methods section and results section (e. g. a supplemental table or figure could be included), respectively.

Moreover, I believe codon volatility in IDRs and ORs should not be compared directly, as amino acid composition and the length of these regions constitute a bias, much as they do for the analysis of S and NS mutation distributions in these regions. The authors have been careful to compare these mutation distributions to simulated “control” data, but do not take this precaution when comparing codon volatilities between IDRs and ORs.

It would be preferable to compare codon volatility in IDRs/ORs to a set of simulated synonymous sequences to see whether codon volatility in these regions deviates from what is expected “randomly” given the codon usage in the potyviral genomes.

A method for computing volatility P-values for whole genes has been presented in Plotkin et al. 2004 (doi: 10.1038/nature02458) and further explained in Plotkin et al. 2006 (doi:10.1093/molbev/msl021) and might be applicable to gene regions (IDRs and ORs) of potyviruses. The authors could explore this possibility, although I am not sure whether this is easily applicable to small genomes like potyvirus genomes. This might yield results different from those presented in the manuscript where no difference between codon volatilities in IDRs and ORs was detected.

Response :  As suggested, we performed a comparison between the natural volatility of PVY, TEV and TuMV genomes with a dataset of 1000 variants randomly generated with nucleotide variations of the same amino acid sequences. We wrote an in house script and uploaded it on Zenodo (ref). The script from Plotkin 2006 was not available. Volatility comparison between IDRs and ORs and the associated comparison with simulated data are provided as supplemental figure S4.  Text (line 149-158 material and methods, and line 349-357) was modified accordingly.

-  In the beginning of the introduction, mutational robustness is defined as “the capacity to accumulate amino acid substitutions without drastic impairment on protein function”. This definition implies that protein function is unaffected by the accumulated mutations, and therefore based on this definition, strictly speaking, mutational robustness should not be invoked unless protein function is assessed.

Response : Introduction (line 51-52), « mutational robustness » was removed.

Elsewhere in the manuscript no protein function/virus phenotype is assessed but the term “mutational robustness” is used (9 occurrences) in a way that suggests the authors are merely considering the first part of their definition “the capacity to accumulate amino acid substitutions”.

For example in the discussion: “In this study, we tested the hypothesis that among these viruses, the accumulation of non-synonymous mutations (NS) which we simply relate here to mutational robustness was greater in the disordered regions of their proteomes than in the ordered regions.”

The verb “relate to”  is a bit vague. Do the authors mean "define here as" ?

Response : Discussion (line 352-353), « robustess » was removed.

It should be at least stated somewhere in the text that only the first condition (the accumulation of amino acid substitutions) is addressed in this manuscript and that whether the mutations are deleterious or not in certain hosts or environmental conditions remains to be evaluated.

Alternatively, the term mutational robustness should be replaced by another term, or defined differently.

Response: Introduction (line 74-76), we define the term « mutational robustness » in the context of the study. We specified as suggested that this do not include the notion of protein function.

- In the Discussion section, the end of the first subsection “Mutational robustness differences between IDRs and ORs” cited below is confusing to me, particularly now that the previous sentences have been removed.

“Although amino acid polymorphism in these regions may participate in potyvirus adaptation, the conservation of intrinsic disorder during evolution is the result primarily of the second process [what is the first process and where is it mentioned in the previous sentences? Now that the sentence “To obtain the diversity observed in IDRs, two successive processes, namely the generation of mutations and their selection are involved.” has been removed from the discussion, this phrase no longer makes sense], a selection pressure dictated by the essential biochemical functions it [what does “it” refer to here ? “it” should be replaced with “they” (these regions). ] performs to ensure virus replication in the host. In any case, the mutational robustness and diversity that arise from the selection of structure-function relationships within IDRs [does this mean there may be strong structure-function relationships in IDRs after all? Why would they necessarily lead to mutational robustness?] is likely to favor the adaptive potential of the virus. It cannot therefore be excluded that IDRs are also selected according to this last criterion [structure function relationships?], even if this hypothesis remains difficult to assess.”

Is the aim simply to state that, despite high amino acid polymorphism in these regions, IDRs are conserved in potyvirus proteomes and that the explanation for this is that they ensure important biochemical functions in the virus cycle of infection?

Response: Discussion (line 391-397) The associated sentence was rephrased as suggested.

- In the Discussion section, the second subsection is dedicated to discussing results from the literature. This is perfectly all right, but it seems a bit odd to me that this should be made into a subsection. Given the length of the discussion, in my opinion, subsections are not necessary.

Response: Subsections within the discussion have been removed accordingly.

Specific comments

See attached pdf file of annotated revised manuscript for specific suggestions and comments.

Response: All the editors corrections have been integrated to the final version of the revised manuscript.